# Prefrontal cortical plasticity during learning of cognitive tasks

Hua Tang [1,2,3,8], Mitchell R. Riley[1,8], Balbir Singh[1,4], Xue-Lian Qi[1], David T. Blake[5] & Christos Constantinidis [1,4,6,7 ✉]

Training in working memory tasks is associated with lasting changes in prefrontal cortical activity. To assess the neural activity changes induced by training, we recorded single units, multi-unit activity (MUA) and local field potentials (LFP) with chronic electrode arrays implanted in the prefrontal cortex of two monkeys, throughout the period they were trained to perform cognitive tasks. Mastering different task phases was associated with distinct changes in neural activity, which included recruitment of larger numbers of neurons, increases or decreases of their firing rate, changes in the correlation structure between neurons, and redistribution of power across LFP frequency bands. In every training phase, changes induced by the actively learned task were also observed in a control task, which remained the same across the training period. Our results reveal how learning to perform cognitive tasks induces plasticity of prefrontal cortical activity, and how activity changes may generalize between tasks.

[1] Department of Neurobiology & Anatomy, Wake Forest School of Medicine, Winston-Salem, NC 27157, USA. [2] Center for Neuropsychiatric Diseases, Institute of Life Science, Nanchang University, Nanchang 330031 Jiangxi, China. [3] Laboratory of Neuropsychology, National Institutes of Mental Health, NIH, Bethesda, MD 20892, USA. [4] Department of Biomedical Engineering, Vanderbilt University, Nashville, TN 37235, USA. [5] Department of Neuroscience and Regenerative Medicine, Medical College of Georgia, Augusta University, Augusta, GA 30912, USA. [6] Neuroscience Program, Vanderbilt University, Nashville, TN 37235, USA. [7] Department of Ophthalmology and Visual Sciences, Vanderbilt University Medical Center, Nashville, TN 37232, USA. [8] These authors contributed equally: Hua Tang, Mitchell R. Riley. ✉email: Christos.Constantinidis.1@vanderbilt.edu

Working memory, the ability to retain and manipulate information over a period of seconds, represents a core component of higher cognitive functions, including control of attention, non-verbal reasoning, and academic performance[1–3]. Working memory ability has been traditionally thought of as an immutable aptitude, but it is now understood that it can be improved by training in working memory tasks[4–6]. The extent over which performance improvement after working memory training generalizes, or transfers, to tasks that were not part of the training has been a matter of debate; some studies have been successful in inducing transfer from one task to another whereas others have not[4–11]. Less contested is the idea that working memory training is beneficial for patients with clinical conditions, including attention deficit hyperactivity disorder (ADHD), traumatic brain injury, and schizophrenia[4,12,13].

The neural basis of transfer has been poorly understood. Human fMRI studies have produced conflicting results about the effects of cognitive training, suggesting overall increases[13–18], or decreases in activity[19–22], or more subtle differences such as changes in network modularity[23,24]. Increases are interpreted as reflecting a higher level of activation or recruitment of a larger cortical area, decreases as suggestive of improvements in efficiency[25,26]. What these correspond to at the level of neural spiking activity and how lasting changes can transfer between tasks has been hitherto unexplained.

We were thus motivated to address the neural effects of training in a spatial working memory task that with neurophysiological recordings in monkeys. Persistent discharges that continue to represent stimulus properties are thought to underlie working memory, though this is a topic of recent debate, as well[27,28]. We standardized the training in discrete phases and tracked neuronal activity with a chronically implanted electrode array throughout several months of training. We were thus able to address changes in neuronal activity as training proceeded and test how neural activity changes were evident across different tasks.

## Results

**Monkeys acquire different elements of cognitive tasks with training.** Two male Rhesus monkeys (*Macaca mulatta*) were initially acclimated with the laboratory and trained to maintain fixation and not respond to stimuli presented on a computer screen. The monkeys were then trained to perform a spatial working memory task, requiring them to maintain fixation, observe two stimuli appearing in sequence separated by delay periods, and to indicate if the two stimuli appeared at the same location or not by selecting one of two choice targets, defined by their shape ("H" or "Diamond" in Fig. 1a–e). The training to acquire and master this task consisted of four phases. First, the monkey was presented with two stimuli in rapid succession and had to indicate if they appeared at the same or different locations by selecting one of two choice targets (Fig. 1b). During this phase, daily sessions involved the presentation of the cue at the right of the fixation point followed by a sample stimulus appearing at either a matching location (right) or a nonmatching location (left), on different days. At this stage, the monkey could simply sample the choice targets, determine which one was rewarded during the block, and repeatedly select it in following trials. In the second phase, the monkey was presented with alternating blocks of match and nonmatch trials, of decreasing block length, until they were randomly interleaved, requiring the monkey to associate the second stimulus location with the corresponding choice target (Fig. 1c). In the third phase, the monkey had to generalize the task to new stimulus locations, appearing at a 3 × 3 grid (Fig. 1d). Finally, an increasing delay period was imposed, placing more demand on working memory (Fig. 1e).

Importantly, visual stimuli were also presented to the monkeys passively, in a control, fixation task every day (Fig. 1a). The sequence of events in the passive trials mirrored the final phase of the active task; a stimulus was presented at a random location, followed by a second stimulus appearing either at a matching or a nonmatching location, separated by 1.5 s delay periods. The critical difference was that no choice targets were presented in the passive task, and the monkey was rewarded at the end of the second delay period merely for maintaining fixation; no response of any kind was required. The monkeys performed this passive fixation task before recordings began, and they continued to perform it in exactly the same fashion at the beginning of each daily session before active task training began. Training proceeded in an adaptive manner, so that the task became progressively harder as the monkeys mastered each element of the task so that overall performance remained approximately constant through the duration of the training (Fig. 1f, g).

**Training increases neuronal activation.** After initial acclimation with the laboratory and before Phase I training began, the animals were implanted with a chronic array of electrodes in their lateral prefrontal cortex (Fig. 2a). The implant comprised an 8 × 8 grid of electrodes, with adjacent electrodes spaced 0.75 mm apart from each other, thus covering an area of 5.25 mm × 5.25 mm. The electrode array was implanted in the dorsolateral prefrontal cortex (dlPFC), with electrode tracks descending in both banks of the principal sulcus (Fig. 2b, c). Local field potentials (LFP) and multi-unit activity (MUA) were recorded from all electrodes, which remained fixed after training began. To sample spiking activity in an unbiased fashion, we set the exact same MUA threshold criterion for all electrodes and sessions, to 3.5 × root mean square (RMS) of the noise level. We were thus able to quantify systematic changes in neural activity as training took place. We identified MUA units with responsiveness to stimuli as those exhibiting a significant elevation of firing rate during either the first stimulus presentation or the delay period following it (see Methods). A total of 4537 responsive MUA units were identified in this fashion across all phases of training and across all electrodes, with a sustained yield of responsive units through the last training phase (Fig. 2d–f). Single neuron recordings were also obtained, after spike sorting of the MUA records. A total of 1093 single units were recorded from the active task and 1065 from the passive task; of those 49.2% were responsive in the active task and 27.2% in the passive task, based on the same criteria.

We next addressed the effects of training on neural activity. Based on experimental and theoretical grounds[29], we hypothesized that a greater proportion of neurons would be activated, and at a higher firing rate. Indeed, training in the active task resulted in a greater population of prefrontal MUAs becoming responsive to the stimuli (Fig. 2d, 1-way ANOVA test; orange bars for MUAs recorded in active task, $F_{3,\ 1017} = 73.32$, $p = 3.78 \times 10^{-43}$; cyan bars for MUAs recorded in passive task, $F_{4,\ 535} = 3.34$, $p = 0.010$), and in a higher mean firing rate generated by single neurons (Fig. 3a–c). Comparison of mean firing rates of responsive single neurons for the best location of each neuron in each training phase, after subtracting baseline activity, revealed a highly significant difference between stages (Fig. 3b; 1-way ANOVA test, $F_{3,531} = 72.4$, $p = 2.87 \times 10^{-39}$ for the cue period, $F_{3,531} = 21.93$, $p = 2.12 \times 10^{-13}$ for the first delay period). These changes in firing rate were also evident in the context of the passive fixation task (Fig. 3d–f), though changes were not always monotonic or as consistent. Firing rates for the best location after subtracting the baseline was significantly different between phases

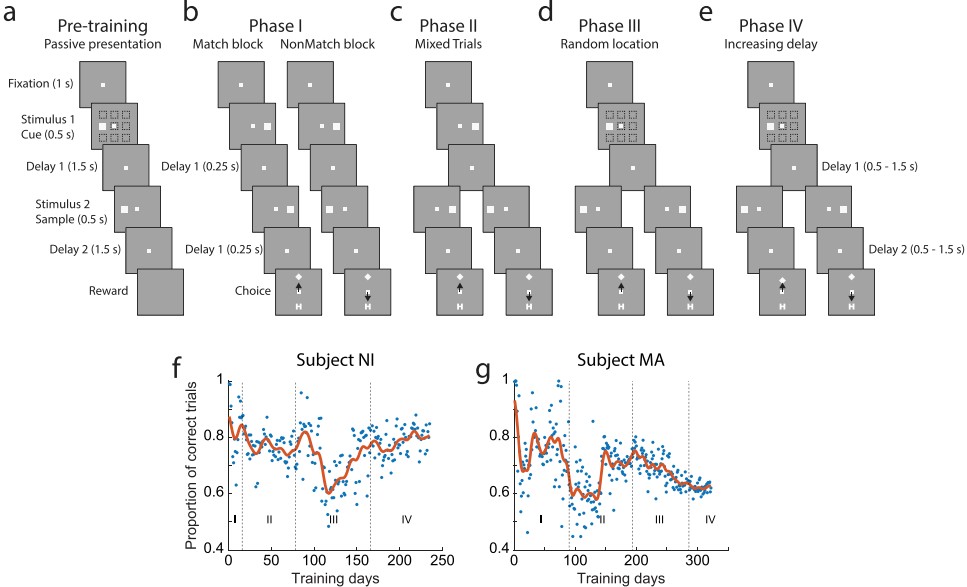

**Fig. 1 Behavioral Training. a–e** Successive frames illustrate the sequence of events in the tasks used in progressing training phases. **a** During the pre-training phase, the monkey only had to fixate while the stimuli were displayed at any one of the nine locations on the screen. **b** In Phase I, a stimulus was always presented to the right, followed by a match stimulus in a block of trials and by a nonmatch stimulus in another block of trials. At the end of the trial, two choice targets appeared, and the monkey had to choose the "Diamond" target in match blocks and the "H" target in nonmatch blocks to get a reward. **c** In Phase II, match and nonmatch trials were mixed in a block. **d** In Phase III, the stimulus location of the first stimulus could vary. **e** In Phase IV, the duration of the delay period increased. The passive stimulus set continued to be presented at the beginning of each session throughout training. **f**, **g** Performance of two monkeys at each daily session. Each dot represents one day's performance. Red lines represent data averaged over 30 days.

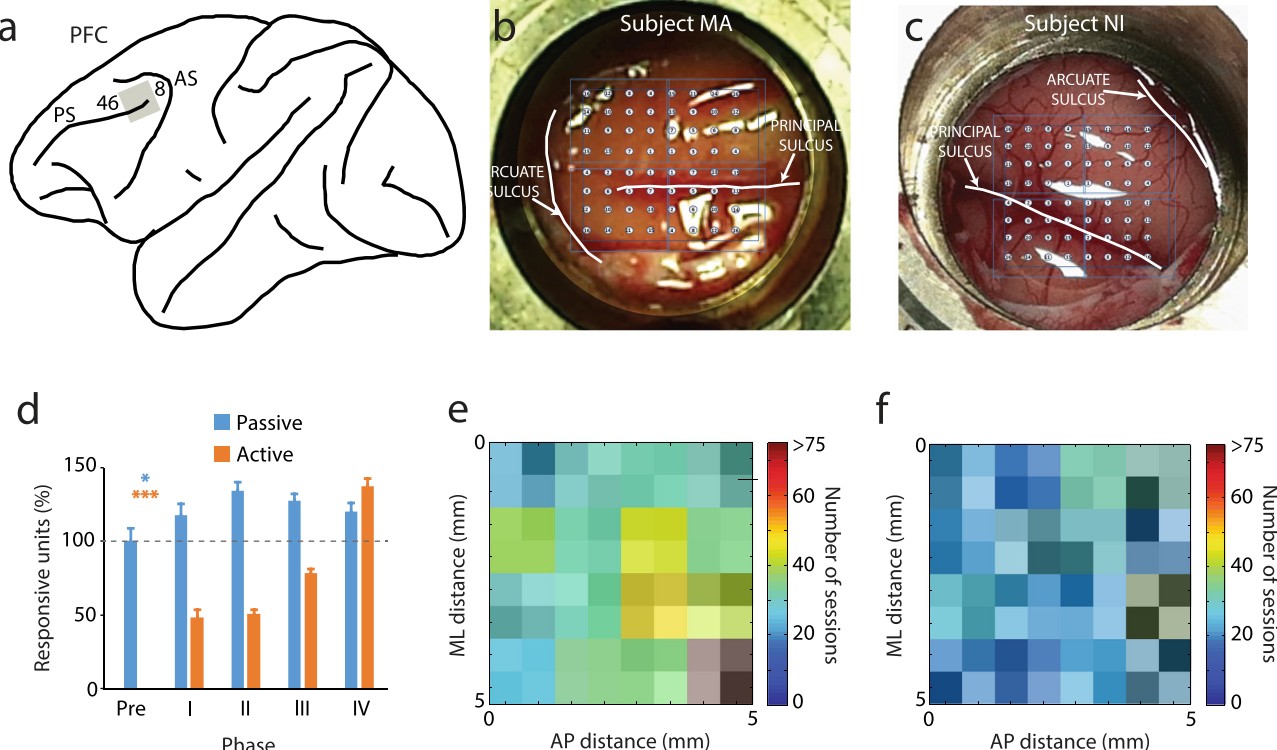

**Fig. 2 Chronic array recordings. a** Schematic diagram of the monkey brain with the approximate location of the recording grid (gray square) indicated relative to prefrontal landmarks: areas 46 and 8, Principal Sulcus (PS) and Arcuate Sulcus (AS). **b** Position of the electrode array in the right prefrontal cortex of monkey MA is indicated relative to the PS and AS. **c** Position of the electrode array in the left hemisphere of monkey NI. **d** Relative numbers of responsive units in each training phase for passive and active tasks. The number of units is shown as a proportion relative to the average unit number of the passive task in the pre-training phase. Data from two subjects, for MA, $n = 1341$ in the passive task, $n = 1150$ in the active task; for NI, $n = 816$ in the passive task, $n = 1230$ in the active task. Error bars represent SEM. Stars indicate significant effects in 1-way ANOVA: *$p < 0.05$, ***$p < 0.001$. Number of sessions for each electrode that showed response to the task, plotted by their location in the array located in monkey MA (**e**) and NI (**f**).

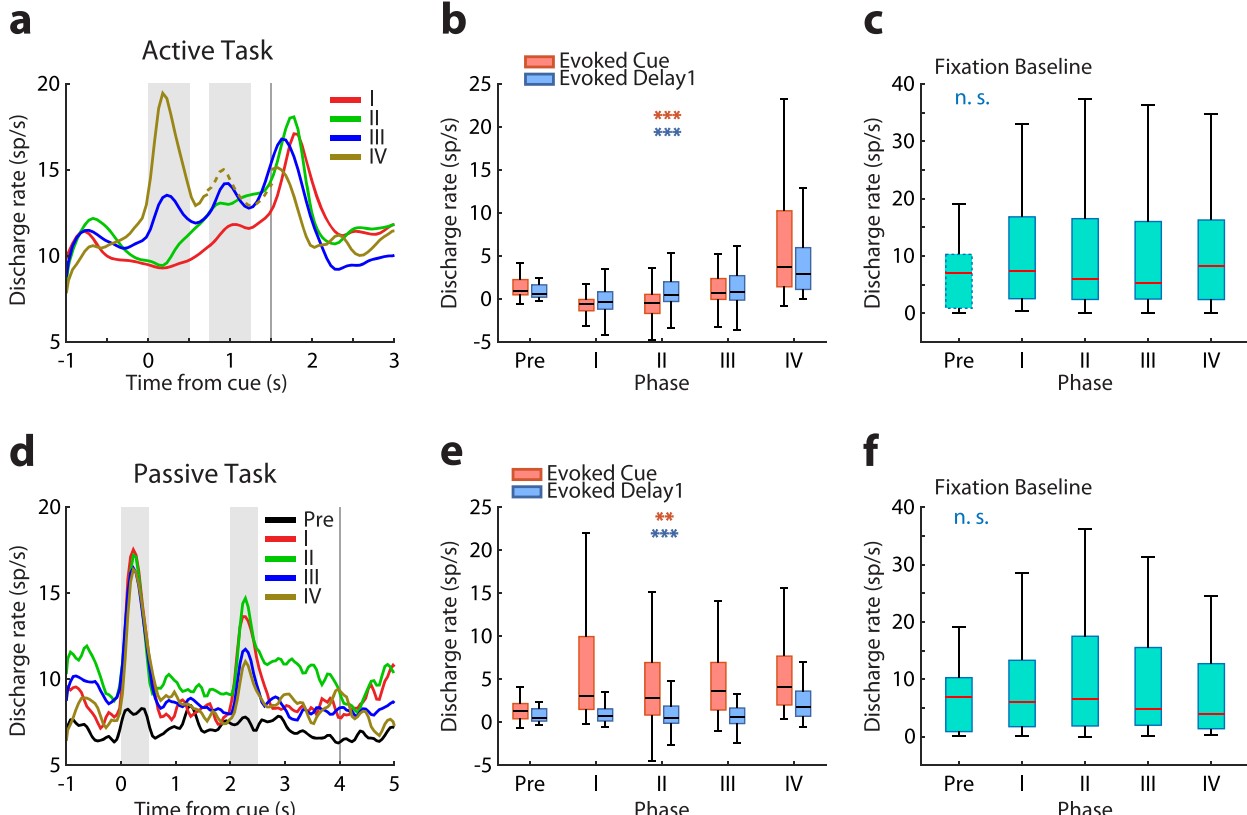

**Fig. 3 Mean firing rate of single neurons at different training phases. a** Population peri-stimulus time histogram (PSTH) of responsive single neurons in the active task (*n* = 538). Best stimulus location for each responsive neuron is used, aligned to the cue presentation. Shaded areas represent the stimulus presentation periods; vertical line, the onset of the choice targets. The delay period was variable in phase IV; only the first 250 ms are indicated (activity followed the second stimulus is plotted in dotted line); the rest of the plot is aligned to the onset of the choice targets. **b** Neuronal activity averaged over the cue and first delay periods after subtracting the baseline is plotted for each of the training phases. Each box indicates the median, first and third quartile, and 1.5x interquartile range of this firing rate relative to baseline. **c** Baseline fixation for the active task. *n* = 17/85/191/170/92 for each phase for the evoked cue, evoked delay1 and fixation baseline. **d** Population PSTH of all responsive units in the passive task (*n* = 290). **e**, **f** Data plotted as in panels b and c, for the passive task. *n* = 17/41/67/97/68 for each phase for the evoked cue, evoked delay1 and fixation baseline. Error bars represent SEM. Stars indicate significant effects in 1-way ANOVA; **\**p* < 0.01, \**\*p* < 0.001, n. s. not significant.

(Fig. 3e; 1-way ANOVA test, $F_{4,285}$ = 4.1, *p* = 0.003 for the cue period, $F_{4, 285}$ = 5.41, *p* = 0.0003 for the first delay period). Even though stimuli were presented exactly in the same fashion every day, prefrontal single neurons generated higher levels of activity after the monkeys had been trained to perform a task. The increase in firing rate in the passive task after training began in the active task was observed in both monkeys (Supplementary Fig. 1), and even though the location of the trained cue in the active task was ipsilateral to the recording array for one monkey and contralateral for the other.

The cumulative effect of a greater population of units being recruited and firing at a higher rate during the passive task as training in the active task progressed could be appreciated when we tracked MUA activity from the same channel over repeated days. Absolute activity in the example channels illustrated in Supplementary Fig. 2 peaked at phase III (when the monkey mastered the full task, in active training sessions practiced later in the day). This increase in firing rate was evident already from the baseline fixation interval, though peak cue and delay period activities also changed during the course of training. It was also important to realize that the firing rate changed continually even within each phase, as the monkey figured out new elements of the task and improved in performance. This can be appreciated when we plotted the MUA firing rate on a day-to-day basis, as training progressed (Fig. 4a–c). This illustration also made evident that a more granular analysis

was necessary to understand the nature of neuronal activity changes during training and how these manifest themselves across tasks.

**Neural effects of acquisition of different task element transfer between tasks**. Training in Phase I required the monkeys for the first time to observe the choice targets and select one as a saccade target, creating associations between sensory stimuli and reward or its omission. We point out that in the pre-training phase, if a monkey responded to any stimulus, the reward was omitted. Trials with cue and match presentations alternated with cue and nonmatch presentations in different sessions. The subject could perform the task by simply ignoring the two first stimulus presentations, waiting until the choice targets appeared, and testing which one of the two was rewarded, then returning to the rewarded target in all subsequent trials of the session. We hypothesized that the significance of these task events would be reflected in neural variables. Indeed, a peak in firing rate (Fig. 3a) was evident at the time the choice targets appeared. Little phasic response was evident during the presentation of the cue and match/nonmatch. However, activity ramped during the time course of the trial, peaking before the appearance of the choice targets, from 9.3 spikes/s in the cue period to 12.7 spikes/s at the onset of targets (Fig. 3a). We also postulated that the active engagement in the task would result in heightened activation of

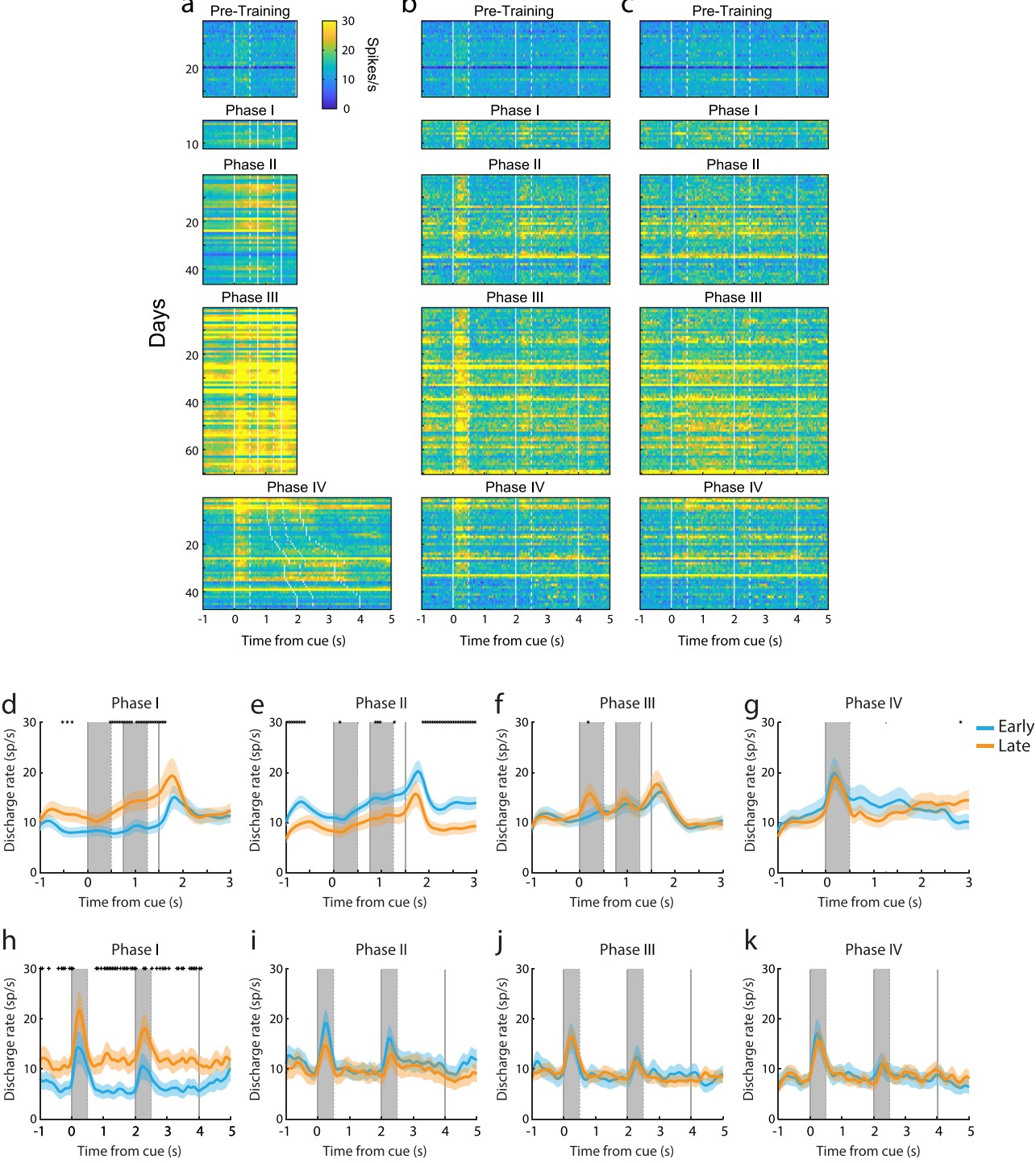

**Fig. 4 Daily responses in the active and passive tasks as the active training progressed.** Activity of MUA units responsive to the active task (**a**) and the passive task (**b**, **c**). Color plot represents the mean firing rate of all responsive MUA units available on that day. Only days with responsive MUA units in both the active and passive tasks were identified are shown. Data are plotted for the best cue location (**a**, **b**) and the best delay period (**c**) activity of the MUA units under study. **d–g** Population PSTH of responsive neurons in the active task ($n = 44/41$, 104/87, 85/85, 47/45 for early and late phases in Fig. 4d–g). **h–k** Population PSTH of responsive neurons in the passive task ($n = 24/17$, 34/33, 39/58, 32/36 for early and late phases in Fig. 4h–k). Shaded zones represent mean ± SEM. The black asterisks indicate a significant difference between the early and late training phases (two-sided t-test, $p < 0.05$).

the prefrontal cortex in the baseline period and during the presentation of the visual stimuli, now in the context of a task. This expectation was also confirmed. During the course of learning the association between choice targets and reward, long-lasting changes in the prefrontal network were observed, which also transferred during the passive task: firing rate during the

execution of the active task increased during the course of training (Fig. 4d, two-sided t-test, $t(134) = 3.07$, $p = 0.003$). The same rate change was also observed in the passive task (Fig. 4h, two-sided t-test, $t(39) = 2.22$, $p = 0.032$).

In Phase II, presentations of both match and nonmatch trials occurred during the same session. At the initial training sessions,

match trials were presented until the subject completed 50 correct responses, and these were followed by nonmatch trials. In this stage, too, the subject could perform the task by ignoring the cue and match/nonmatch stimulus and relying on reversal learning of the rewarded choice target. However, as the blocks of match and nonmatch trials became shorter, and eventually fully randomized, the subject could only perform the task by becoming aware that the "Diamond" choice target was associated with the match stimulus and the "H" choice target with the nonmatch. We note that throughout phase II, the monkeys could still perform the task by essentially ignoring the cue (first stimulus), since it always appeared at the same location. The significance of these task events was also reflected in neural variables. Little response was present during the cue period (Fig. 3a, b), but firing rate further accelerated during the second stimulus presentation prior to the saccade, reaching 15.4 spikes/s at the onset of choice targets (Fig. 3a). Based on experimental results in the sensory cortex, we hypothesized that this training would induce a transient, non-selective increase in responsiveness[30], which we expected would be reflected in baseline and stimulus-driven firing rate. We indeed observed changes in the prefrontal network, which also transferred during the passive task: firing rate during the execution of the active task initially increased relative to Phase I, but then decreased again during the course of training (Fig. 4e, two-sided $t$-test, $t(298) = 2.12$, $p = 0.035$). A parallel pattern of rate changes was not observed in the passive task (Fig. 4i, two-sided $t$-test, $t(65) = 0.49$, $p = 0.624$). These effects were evident in the day-to-day changes (Fig. 4a–c). We note that different neurons were responsive in the active and passive task; these changes reflected overall changes in responsiveness across the prefrontal network, rather than sampling of neurons with lower or higher activity at different recording dates.

Training in Phase III required the subjects to generalize across multiple cue locations. In order to perform the task, the monkeys now needed to observe and remember the location of the cue and compare it with the location of the second stimulus in order to determine if that was a match or not and plan the appropriate response. We anticipated that expanding the range of stimulus locations would produce further changes in neural recruitment. Indeed, responses to the cue stimulus, which now became essential for the task, increased greatly (Fig. 3a). However, by virtue of presenting the cue at multiple locations, more neurons had a chance of being activated, whereas no such change occurred in the presentation of stimuli in the passive task. Progression of training in this phase was characterized by stability in other aspects of neural activity; no change in baseline firing rate was evident between early and late training phases (Fig. 4f, two-sided $t$-test, $t(234) = 0.73$, $p = 0.468$) and these negative findings were also shared in the passive tasks (Fig. 4j, two-sided $t$-test, $t(95) = 0.02$, $p = 0.987$).

Phase IV amplified the working memory demand of the task, as the duration of each of the two delay periods in the trial progressively increased from 0.25 s to 1.25 s. The most salient change in neural activity was the increase in firing rate during the first delay period relative to the baseline (Fig. 3b). As the timing of task events changed for the first time during training, the ramping of activity after the cue presentation also disappeared (Fig. 3a). This change occurred rapidly, as soon as the delay period began increasing in the active task (Supplementary Fig. 3). The elimination of ramping activity has been reported in working memory tasks that randomize the delay period compared to versions of the task with a fixed delay period[31]. As was the case in phase III, some of these changes were transient. The absolute level of activity declined later in the phase (this is evident in Fig. 4a as well). We have recently reported an analogous phenomenon of working memory activity becoming more distributed across a larger population of neurons, while individual activity decreases, in an experiment relying on single-neuron recordings at early and late phases of a working memory task with multiple stimuli[26]. Increasing the delay period of the active task also induced long-lasting changes in the prefrontal network, which were evident in recordings during the passive task: Increased delay period relative to baseline was now evident in passive recordings, the only phase in which this occurred (Fig. 3e, two-sided $t$-test, $t(67) = 7.17$, $p = 7.49 \times 10^{-10}$). A decrease in the baseline firing rate was also observed in the passive task (Fig. 3f). The common trajectories of changes in the activity of passive and active tasks could also be appreciated in the day-to-day firing rate changes (Supplementary Fig. 4).

In addition to analyzing responses to the best location of each neuron in the passive task based on the phases of task learning, it was also important to examine how responses to the same location changed as a function of experiencing these stimuli in the context of the task. The first two phases of the active task involved training with stimuli always presented at the same two locations, in the left and right of the screen, followed by choice targets at orthogonal locations, at the top and bottom. Responses to stimuli at other locations in the passive task were altered during this period even though the monkey had not actively been trained with them yet. Such an example is shown for Supplementary Fig. 5, always tracking responses recorded in the passive task, following cue presentation at the same (lower right) location. A 1-way ANOVA test indicated a significantly different firing rate at the four training phases (Supplementary Fig. 5; $F_{3,223} = 5.89$, $p = 6.97 \times 10^{-4}$ for the cue period, $F_{3,223} = 7.49$, $p = 8.43 \times 10^{-5}$ for the first delay period). In the middle of phase III, the lower-right location became the site of one of the two choice targets in the active task, when the cue and match stimuli were first presented in the locations diagonal to it, in the upper-right or lower-right location. This was also associated with a large increase in firing rate for the presentation of the cue stimulus in the lower-right location in the passive task. Finally, when the monkey was exposed to stimuli appearing at the lower right location as cues that needed to be remembered in phases III and IV, responses to stimuli at that location actually declined in the passive task. These results suggest that changes in response were not tethered to the specific stimulus being used in the context of the active task but were more general, as the network was altered during training.

**Decoding of task variables improves as stimulus decoding remains stable.** To understand how training affected the type of information represented in neural populations, we performed a decoding analysis at different phases, relying on Multi-Unit Activity records. The decoder readily extracted the location of the first and second stimulus from the passive and active tasks (Fig. 5a–c). On the other hand, the match or nonmatch status of the second stimulus could barely be decoded from the passive task with above chance accuracy at any phase of training; mean decoding accuracy during sample period was 0.51, 0.51, 0.53, 0.51, and 0.50, for pre-training phase and the four training phases respectively (Fig. 5d). This was in stark contrast with match-nonmatch information being readily decoded from the active task; mean decoding accuracy in the sample period was 0.58, 0.67, and 0.58 for phases II, III, and IV respectively – though we should note that in early phases match and nonmatch stimuli were presented in blocks and differential responses might represent anticipated responses in some extent (Fig. 5e, f). To ensure that this selectivity for the matching or nonmatching status of a stimulus was not driven entirely from the initial, left-right, set of locations, we also performed this analysis separately for each pair of match-nonmatch locations (Supplementary Fig. 6). Robust decoding was present for all locations

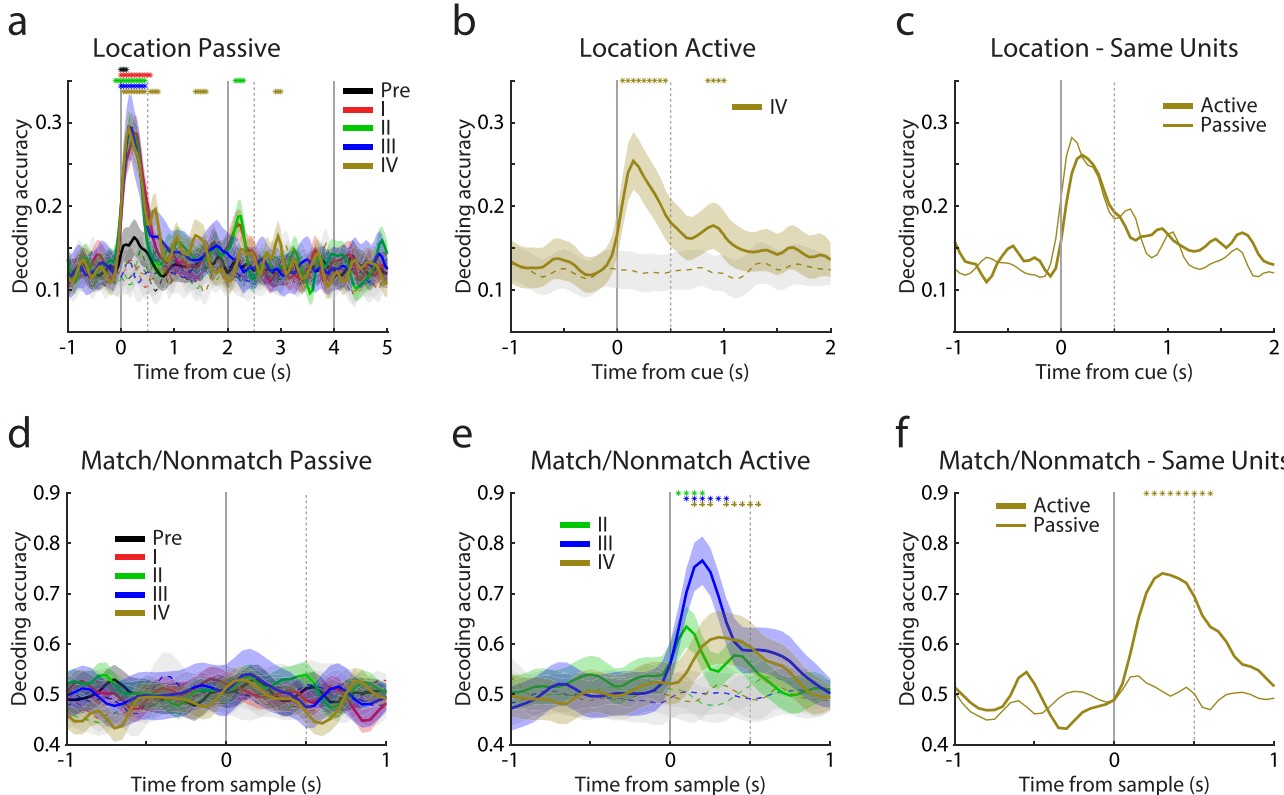

**Fig. 5 Decoder analysis. a**, **b** Accuracy of decoding stimulus locations based on multi-unit activity pooled from the passive task (**c**), and the active task (**d**), separately for each training phase. Only phase IV is included in the active task, as only in this the monkey has been trained to distinguish all locations. **d**, **e** Accuracy of decoding the match or nonmatch status of the second stimulus based on multi-unit activity pooled from the passive task (**a**) the active task (**b**), plotted separately for each training phase. Only phases II-IV are included in the active task, as only in these the monkey has been trained to distinguish between match and nonmatch choices. The colored asterisks indicate a significant difference (two-sided Z-test; $p < 0.05$) between the corresponding area and the shuffled data. Dash lines represent the mean decoding accuracy of shuffled data. Shaded zones represent mean ± SD. Pseudo-populations of 200 randomly selected MUAs were used in each stage in Fig. **a**, **b**, **d**, **e**; results were averaged from 100 resample, $n = 100$. **c**, **f** Decoding of stimulus location information and match-nonmatch information from the exact same neurons ($n = 72$) in the passive and active tasks, without resampling. The asterisks indicate a significant difference (two-sided Z-test; $p < 0.05$) between the passive and active tasks, across cross-validations.

introduced during phase III of training (Supplementary Fig. 6b–d). As an alternative means of quantifying changing selectivity for stimulus information across tasks, we performed an ANOVA test for location and matching status of a stimulus (Supplementary Fig. 7). This largely confirmed the decoding results: selectivity for the matching or nonmatching status of the stimulus was much greater in the active than the passive task. These negative findings provide assurance that the changes we did observe in the passive task represent neural effects that generalize across tasks, rather than implicit execution of the active working memory task even during passive fixation.

To formally identify the types of information represented in the activity of the passive task, we used demixed Principal Component Analysis, which decomposes the matrix of neural activity across neurons based on task and stimulus information[32]. The representation of stimulus locations, match or nonmatch status of a trial, and invariant components remained fairly unchanged in the activity of neurons during the execution of the passive task across the training phases (Supplementary Fig. 8). Most importantly, decision components, which represent information about the match and nonmatch status of the second stimulus were virtually absent in the passive task across all training phases.

**Training in some task elements decreases noise correlation.** In order to understand the changes in the connectivity structure of the network as training took place, we computed spike-count correlation (also known as noise correlation) between pairs of single neurons recorded simultaneously in the same sessions[33]. A total of 2685 pairs of single neurons were used in this analysis. We relied on spikes recorded during the baseline fixation interval of the task, which was identical across tasks and training phases. Across all conditions tested, there was a strong dependence of noise correlation on the lateral distance between the electrodes from which the neurons were recorded (Fig. 6), in agreement with prior studies[34]. Additionally, the noise correlation computed during the passive task was consistently higher than in the active task, also in agreement with prior studies suggesting that factors such as attention and arousal decrease noise correlation[35]. Importantly, noise correlation differed systematically between training phases. The predominant effect of training was a decrease in noise correlation, which confirmed prior studies comparing naïve with fully trained monkeys[36]. However, this change occurred mostly in training phases III and IV (Fig. 6). An Analysis of Covariance comparing noise correlation at different training phases of the active task indicated a significant difference between phases, after accounting for distance, which was used as a covariate ($F_{3,2598} = 10.83$, $p = 4.5 \times 10^{-7}$). Essentially the same effect was present when we repeated the same analysis in noise correlation computed in the passive task ($F_{3,2555} = 9.37$, $p = 3.7 \times 10^{-6}$). Focusing exclusively on noise correlation computed from neurons recorded from the same electrode, where the largest samples were available, confirmed these findings (Fig. 6i).

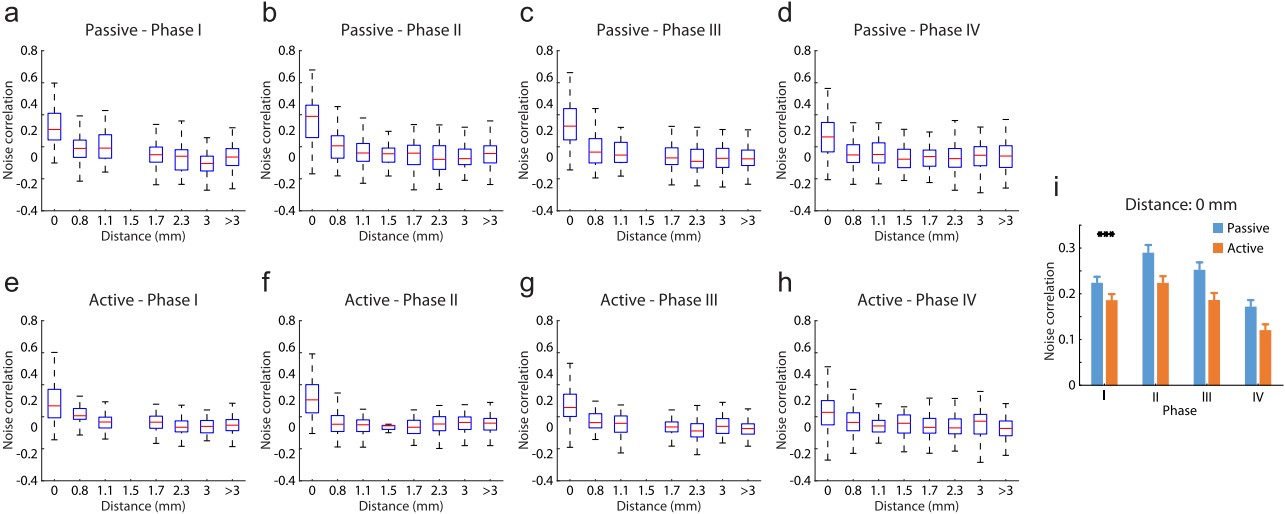

**Fig. 6 Noise correlation. a–d** Box plot of spike-count correlation values (noise correlation) for pairs of neurons ($n = 2576$) recorded in the passive task, as a function of distance between electrodes, at different training phases. Each box indicates the median, first and third quartile, and 1.5x interquartile range of noise correlation values computed from pairs of neurons at the distance indicated in the abscissa. **e–h**. As in **a–d**, for spike-count correlation values ($n = 2603$) in the active task. **i** The average correlation values of the pairs of neurons recorded in the same electrodes ($n = 477$ and 474 for passive and active tasks). Error bars represent SEM. Stars indicate significant effects 2-way ANOVA; ***$p < 0.001$ for tasks and phases.

A 2-way ANOVA using training phase and passive or active task as factors indicated highly significant effects of both phase (Fig. 6i; $F_{3,963} = 21.51$, $p = 1.7 \times 10^{-13}$) and task (Fig. 6i; $F_{1,963} = 28.39$, $p = 1.4 \times 10^{-17}$). These results suggest that some elements of task training produce lasting changes in the underlying neural circuitry, which are reflected in the correlated firing of prefrontal neurons, and these changes too transfer between the active and passive tasks.

**Training reduces LFP beta power.** Lastly, we examined changes in LFP power spectra as a result of training. Theoretical and experimental studies suggest that improved working memory maintenance is associated with decreased power in the beta-frequency band and increased power in the gamma band[37–40]. We, therefore, wished to test the hypothesis that training would produce overall decreases in beta power and increases in gamma. To compare results from different electrodes and sessions, we first normalized the power of each trial to the baseline, fixation period prior to the cue. We then compared power changes during the trial, relative to this baseline. In partial agreement with our hypothesis, training-induced systematic changes in power, the most salient of which was a progressive decrease in power in the beta/low-gamma frequency zone of 20–45 Hz (hereafter referred to as beta, for simplicity) during the cue presentation period in successive active training phases (Fig. 7a). Averaging beta power over the entire cue period revealed a highly significant difference between phases (1-way ANOVA comparing beta power in daily sessions grouped in four training phases, $F_{3,889} = 113.8$, $p = 2.27 \times 10^{-62}$). A concomitant increase in alpha-frequency power (8–14 Hz) was also observed ($F_{3,889} = 94.0$, $p = 7.47 \times 10^{-53}$). High gamma (46–70 Hz) power was less diagnostic of the training progression but generally decreased, contrary to our initial hypothesis. Importantly, those global changes in beta and alpha power were also present in the passive-fixation task (Fig. 7b), which the monkeys continued to be exposed daily, at the beginning of each session before training in the active task began. Although the passive task stimuli never changed, we observed a significant decrease in beta power across successive phases, considering the pre-training phase as well (1-way ANOVA, $F_{4,446} = 33.8$, $p = 1.28 \times 10^{-24}$), and a relative increase in alpha power ($F_{4,446} = 18.0$, $p = 1.02 \times 10^{-13}$). The decrease of beta power/

increase of alpha power across training phases that transferred into the passive task was observed in both monkeys (Supplementary Fig. 9). The effects were essentially identical when we performed LFP analysis only in electrodes from which single neurons were recorded, to ensure that changes detected were not the result of some electrodes becoming inactive (Supplementary Fig. 10); this possibility was remote, in any case, since the dominant effect seen in neurophysiological recordings was an increase in activity.

## Discussion
It has been recently recognized that working memory ability is malleable and can be increased by using computerized training[4–6]. After such training, some performance improvements generalize between tasks by improving not only for the trained tasks but also for tasks that were not part of the training[5,8–10,41–44]. Our study tracked the changes in neural activity that occurred after training in the task being trained and in a control task that remained the same (passive task). This approach allowed us a window on the changes of the prefrontal circuitry as the result of such training-induced plasticity. Across four learning phases that required mastery of different conceptual elements and induced qualitatively distinct changes in neural activity, we consistently observed that neural changes in the prefrontal network through training in the active task were also evident in the passive task. Changes of neuronal activation in the active task included increases in the percentage of units that were responsive to any aspects of the task and stimuli, increases in the mean firing rate of responsive neurons, decreases in noise correlation, and increases in high beta/low gamma LFP power, in agreement with changes previously documented in single-electrode studies comparing different populations of neurons, in naïve and fully trained animals[26,45–48], or during the course of a daily training session, when a specific stimulus is associated with reward[49,50]. Both increases and decreases in activity observed in the active task transferred to the passive task, as did null results (e.g., no baseline activity change during the course of Phase III). Artificial neural networks have provided a framework for understanding transfer learning: a network trained on one task produces changes in connection weights in the hidden layers of the network, which when probed with a different task generate

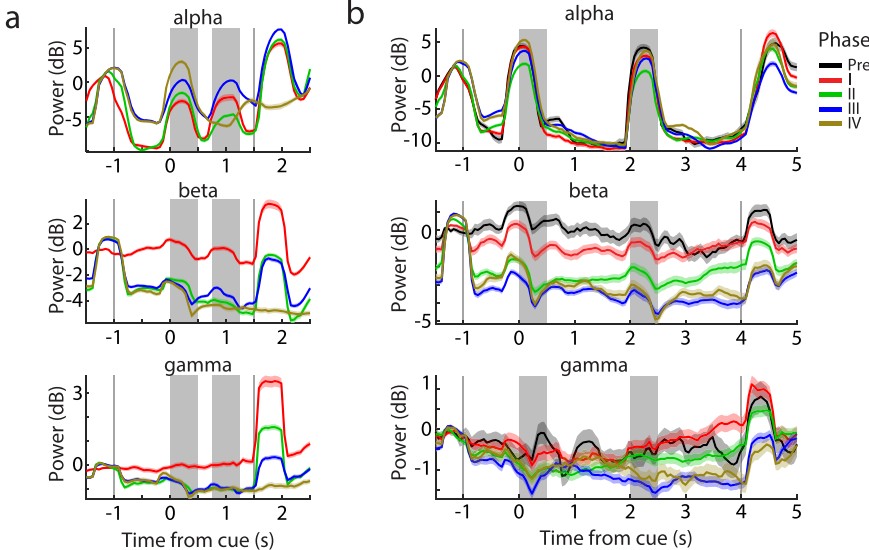

**Fig. 7 LFP analysis. a** Time course of power at discrete frequency bands and different training phases of the active task: alpha (8–14 Hz), beta (20–45 Hz), gamma (46–70 Hz). **b** Time course of power in the same frequency bands for the passive task, as training progressed in the passive task. Shaded areas represent the stimulus presentation periods. Shaded zones represent mean ± SEM.

training-dependent output[51]. We now document the neural equivalent of this process, as learning takes place. However, not all changes that we observed in the active task after training were also present in the passive task. Most importantly, information about whether the second stimulus was a match or a nonmatch was much more pronounced in the active task, throughout training. The results suggest that learning induced long-lasting changes that can be probed with the passive task, however execution of a cognitive task is also characterized by unique information that neurons represent dynamically in the course of a trial, in agreement with previous studies that have shown context-dependent information representation[52].

Recordings from a chronic array of electrodes in the prefrontal cortex throughout training enabled us to monitor changes in neuronal activity in the active task itself. Prefrontal activity is well known to exhibit rapid plasticity in the context of a trained task, for example, during the course of a daily training session, when a specific stimulus is associated with reward[49], or during reversal learning[50]. Previous studies comparing activity sampled from different sites, before training began and after it was completed, emphasized increased activity and recruitment of more neurons after subjects are trained in working memory tasks[46–48]. Learning the rules of the task, but also increased effort and arousal associated with executing the task itself are thought to modulate neuronal activity[46,53]. Our results confirmed these observations, and determined that the number of units activated after task training may have even been underestimated in single electrode studies. On the other hand, we found that many changes associated with different elements of task acquisition were transient or moved in opposite directions at different training phases (Figs. 4 and 5).

Working memory is thought to be mediated by persistent activity generated during the delay interval of working memory tasks, though this has been a matter of debate in recent years[27,28,54–57]. The persistent-activity model of spatial working memory posits that the appearance of a stimulus generates activity that is maintained during the delay period, but may drift with time[58,59]. The location recalled by the subject is precisely determined by the drift of the delay period activity. In this context, working memory training is thought to rely on strengthening network connections between neurons that generate

persistent activity, by virtue of recruiting more neurons during the delay period of the task; by achieving greater discharge rates during the delay period; and by realizing lower variability in firing rate from trial to trial[46,47,60]. Such changes in discharge patterns suggest enduring changes in the prefrontal circuitry after training, which would suggest that the excitability of prefrontal neurons and the ability to generate persistent activity is lastingly altered following training. Our present results are consistent with this interpretation. When probed with passively presented stimuli, larger populations of prefrontal neurons were shown to be active and to achieve higher firing rates even though it was not necessary to process or maintain these stimuli in memory for the requirements of the passive task. Such changes would also be expected to strengthen neuronal responses to other tasks that rely on working memory and maintenance of spatial information in mind. Our results provide a framework for probing such changes in future studies.

Evidence from EEG studies in humans most often associates working memory maintenance with increased gamma power[61] and recent models of working memory emphasize increase of gamma power at times of active memory maintenance[37–39]. However, an increase of power in high beta and low gamma frequency, e.g., in the 24–60 Hz range has also been reported in working memory tasks[62–64], referred to as "beta2"[64]. Guided by these models, we tested for systematic changes in LFP power, and we indeed found consistent decreases in high beta – low gamma power at successive phases of training. High gamma power generally declined, although we should note that working memory demands were maximized only at the very last stages of training. Regardless of the underlying mechanisms that brought about these changes at the level of beta-frequency LFP power, these also transferred to the passive task.

An important consideration for the interpretation of the findings is whether the effects observed in the passive task were the consequence of monkeys mentally "performing" the active task even when presented with stimuli passively, which would also imply increased attention and arousal during the passive task. This possibility is unlikely for multiple reasons: Blocks of trials of the passive task were presented in exactly the same routine fashion, at the beginning of the session every day. The passive task did not involve target stimuli at the end of the trial,

allowing the monkeys to realize that no choice was required, from the first trial of the block. The timing of stimulus presentation differed between passive and active tasks, at least through the first three phases of training (until the duration of the delay period was increased), again making the two tasks appear very different. The first two phases of the active task involved training with stimuli always presented at the same two locations, in the left and right of the screen. Yet, responses in the passive task were altered during this period, even for stimuli that the monkey had not actively been trained with yet. Nor was the monkey able to easily generalize performance of the active task with stimuli appearing at other locations; the entire duration of phase III training was devoted precisely to this purpose. Information about the match or nonmatch status of stimuli, on which decisions are based, and which differs in correct and error trials of active working memory tasks[65,66], was also minimal in the passive task. This instance of negative control notwithstanding, increases in neural activity that were present in both the active and passive task would also be expected to strengthen neuronal responses to other tasks that rely on processing of visual spatial information and maintenance in working memory. Our results provide a framework for probing such changes in future studies.

## Methods

**Subjects**. Two male, rhesus monkeys (*Macaca mulatta*) weighing 8–9 kg were used in this study. All experimental procedures followed guidelines by the U.S. *Public Health Service Policy on Humane Care and Use of Laboratory Animals* and the National Research Council's *Guide for the Care and Use of Laboratory Animals* and were reviewed and approved by the Wake Forest University Institutional Animal Care and Use Committee.

**Surgery and neurophysiology**. The monkeys were initially acclimated with the laboratory and trained to maintain fixation on a white dot while visual stimuli appeared on the screen. After this initial stage of training was complete, the monkeys were implanted with a chronic array of electrodes in their lateral prefrontal cortex. The chronic implant was designed in-house[67] and encompassed 64 parylene-c insulated, Iridium electrodes. The implant comprised an 8 × 8 grid of electrodes, with adjacent electrodes spaced 0.75 mm apart from each other, thus covering an area of 5.25 mm × 5.25 mm. The electrode array targeted the dlPFC, with electrode tracks descending in both banks of the principal sulcus (Fig. 2a). The position of the array was determined based on magnetic resonance imaging (MRI) and verified during the implantation surgery. The electrodes could be advanced into the cortex independently of each other and electrode depths were repeatedly adjusted to optimize placements, up to 5 mm (in the banks of suci), over a period of several weeks. Once electrode positioning was finalized, task training and neurophysiological recordings from the array commenced. Neuronal data from each electrode were recorded throughout the training. Multi-unit data were collected from each electrode from areas 8a and 46 of the dlPFC, using an unbiased spike selection procedure. The threshold for spike acquisition was set at 3.5 × RMS of the baseline signal, for each electrode, each day. The electrical signal from each electrode was amplified, bandpass filtered between 500 Hz and 8 kHz, and recorded and sampled at 30 kHz using a Cerberus system (Blackrock Microsystems, Salt Lake City, UT).

**Behavioral tasks**. The monkeys faced a computer monitor 60 cm away in a dark room with their head fixed. Monkeys were trained to hold their gaze on a 0.2° fixation target displayed on a computer monitor. Visual stimuli were then presented on the screen while eye position was monitored via an infrared eye tracking system (model RK-716; ISCAN, Burlington, MA). Eye position was sampled at 240 Hz, digitized, and recorded. The stimuli where 2° squares that appeared randomly in one of 9 locations arranged on a 3 × 3 grid with 10° spacing between stimuli. Correct completion of a trial resulted in delivery of a liquid reward. Behavioral control was implemented with a custom-designed software system. Visual stimuli display, monitoring of eye position, and the synchronization of stimuli with neurophysiological data were performed with in-house software[68] implemented in the MATLAB environment (Mathworks, Natick, MA), and utilizing the psychophysics toolbox[69].

The monkeys were trained in a Match/Nonmatch task involving four phases. The monkeys were then trained to perform a spatial working memory task, requiring them to maintain fixation, observe two stimuli appearing in sequence separated by delay periods, and to indicate if the two stimuli appeared at the same location or not by making an eye movement to one of two choice targets (Fig. 1). The training could be broken down into four phases. The first phase of training involved training the monkeys to make an eye movement to one of two choice

targets and determining that only one of them is rewarded (Fig. 1b). The phase began with the monkeys being exposed to match trials, requiring an eye movement to the "Diamond" choice target. The first stimulus appeared always at the same location (to the right of fixation), followed by a very brief delay period (0.25 s) and a second presentation of the stimulus at the same location. After the second delay period, the two choice targets appeared with the fixation point turning off, either above or below the fixation point, but randomly switching between trials. In the absence of the fixation target, the monkeys quickly foveated one of the choice targets, and they learned through trial and error that the "Diamond" choice target was rewarded. On a subsequent training day, nonmatch trials were introduced. Now the first stimulus appeared at the right location, but it was followed by a nonmatch stimulus. When the choice targets appeared at the end of the trial, it was the "H" shape that was rewarded. The monkeys quickly reversed and saccaded to the "H" choice target. Phase I of training involved delivering match and nonmatch trials in blocks with decreasing numbers of trials before alternating.

Phase II involved randomly interleaving match and nonmatch trials (Fig. 1c). Through this process, the monkeys eventually associated the concept of "match" with the "Diamond" and "nonmatch" with the "H" shape. Phase II concluded when the monkeys were able to perform the task at 75% correct. This was the most challenging phase of training.

So far in training, the cue stimulus always appeared at the same location. Phase III involved the generalization of stimulus location (Fig. 1d). The first stimulus appeared at a new location, followed by a second stimulus at the same or a different location (diametric, except for the center cue which was followed by nonmatch at an adjacent location). Choice targets appeared orthogonal to the axis defined by these possible stimulus locations. We used the upper-right location as cue first, followed by a match in the same location or a nonmatch at the lower-left location. Once the monkey could do the task with this cue, then the upper-left location was introduced. The monkeys were exposed to the rest of the locations in sequence, with the central location of the grid used as cue last. To facilitate learning, whenever a new location was introduced, we relied again on blocks of match and nonmatch trials. To ensure that the monkeys did not "forget" the previous location, they continued to practice these, and every time a new location was added, randomized trials involving all trained locations were interleaved together. The monkeys were able to progress much faster through this stage, though they did not automatically generalize when a new location was introduced. Some practice was necessary to determine what the appropriate choice was for match and nonmatch stimuli appearing at these novel locations.

The final phase of training, Phase IV, involved progressively increasing the delay period duration. Both delays period between the first and second stimulus, and between the second stimulus and choice targets increased in tandem. Durations varied from 0.25 s to 1.5 s.

At the onset of the working memory task training, the monkeys were already able to maintain fixation, and had already been exposed to the visual stimuli that would eventually be incorporated in the task (white squares, appearing at one of nine locations). The timing of the stimulus presentation mirrored the final phase of the task (Fig. 1a). The only difference was that the choice stimuli were presented at the end of the trial, and the monkeys were rewarded for maintaining fixation after the second delay period. An initial set of recordings was obtained from the chronic array at this phase, providing a baseline of neuronal activity prior to the task training. Additionally, the passive presentation of stimuli continued throughout training; the first block of trials presented every day involved the exact same passive stimulus presentation. Thus, monkeys were aware that they did not need to perform a working memory task.

**LFP analysis**. We used the FieldTrip toolbox[70] for preprocessing analysis and the Chronux package[71] for time-frequency analysis. A bandpass filter (0.5–200 Hz) was first used. We removed line power (60 Hz) from each electrode and trial, if present. We used a generalized linear model to identify electrodes with variance outliers, and we omitted them from the analysis. Therefore, the number of electrodes that were averaged varied from 45–60 in each trial. We then used a multi-taper method to perform a power spectrum analysis of LFP. Power spectra were constructed from all trials and electrodes in each session and then averaged across sessions after subtracting the mean power of the baseline fixation period at each frequency. We then compared the LFP power at each frequency between the control and simulation conditions. We also analyzed the LFP power at different frequency bands defined as alpha (8–14 Hz), beta (20–45 Hz) and gamma (46–70 Hz). Line-plots were constructed based on average and standard deviation across sessions (treating one session as one observation). One-way ANOVA was used to compare LFP power between phases, at each frequency band.

**Spiking data analysis**. All data analysis was implemented with the MATLAB computational environment (Mathworks 2019, Natick, MA). We identified MUAs that were responsive to the task and informative about the stimuli as those whose mean firing rate to the different stimulus conditions were significantly different from each other, determined by 1-way ANOVA ($p < 0.05$). The ANOVA was performed for the firing rate averaged across the entire cue period, and the first delay period and compared across available cue locations (typically 9). For task conditions that involved only one cue location (active task, Phase I and II), responsive neurons were identified as those with firing significantly exceeding the

fixation period firing rate (paired $t$-test, $p < 0.05$) between either the first stimulus presentation or the first delay period. We additionally required a minimum 10% firing rate increase during the stimulus presentation over the fixation interval to avoid false positives. Responsive single neurons were determined in the same way as MUA, but without requiring the 10% proportional increase of activity. Firing rate analyses presented here relied on data from correct trials. For each neuron, we identified the cue location that elicited the best response during the cue presentation period, and during the first delay period, determined independently. Activity of the best location in each day, which was defined by the maximum activity in the cue or first delay periods, was shown in heat maps. Daily responses were evaluated by calculating the average firing rate among all selective sites recorded. To compare active and passive conditions, data from responsive neurons recorded in the active and passive conditions were plotted.

Spike count correlation (also known as noise correlation) was computed for pairs of neurons recorded simultaneously. For the analysis presented here, we relied exclusively on the 1 s fixation period that preceded the stimulus appearance, in either the active or passive task. Noise correlation is the Pearson correlation coefficient between these firing rate values[36].

Decoding analysis was carried on the stimulus direction and decision type (i.e., match or nonmatch) factors. We only relied on the eight peripheral locations for this analysis, since the center location never appeared as a nonmatch. Therefore, chance performance for stimulus location decoding was 12.5% and for decision (match or nonmatch) decoding was 50%. The analysis was carried out using the Neural Decoding Toolbox[72,73]. The decoding accuracy of each neuron population was evaluated in 500 ms bins, advanced in 50 ms increments. In each training phase, for both passive and active tasks, pseudo-populations of 200 randomly selected MUAs were used. Trials from these pseudo-populations were randomly split into training and test sets using 10-fold cross-validation. The procedure was repeated 5 times using a different test split each time. For each cross-validation split, all neuron's firing rates were z-score normalized based on the means and standard deviations calculated using data from the training set. This procedure was repeated over 100 resample runs, where different random pseudo-populations and training and test splits were created on each run, resulting in a total of 500 samples for each comparison. A z-test was applied to test significant differences between the actual results and the shuffled results. The shuffled results were calculated with the same neurons and trials used in the actual data, but with the trial order shuffled across different conditions. If the mean of the decoding value achieved by the actual data was located outside the 95% confidence interval of the shuffled data, the neuron population was considered to exhibit significant decoding ability for the condition under study.

We performed Demixed Principal Component Analysis (dPCA)[32], which decomposes population activity into the stimulus components (8 peripheral stimulus locations, excluding the foveal location) and the decision components (match or nonmatch). The method treats the responses of each neuron to one type of stimulus condition as one dimension and then performs dimensionality reduction to determine components that correspond to stimulus and task variables.

A sliding window 3-way ANOVA was performed to examine the encoding of task variables[74], with 200 ms bins and 50 ms steps. Factors, including the location of first and second stimuli, and decision type (match or nonmatch) were used in the model. The two-sided binomial test was applied to test whether the fraction of responsive neurons, on each time point, is significantly above chance level (5%).

**Reporting summary**. Further information on research design is available in the Nature Research Reporting Summary linked to this article.

## Data availability

The data that support the findings of this study are available from https://data.mendeley.com/datasets/fhx5s7zxg7/2. Citation: Tang, Hua; Constantinidis, Christos (2021), "Dataset for studying prefrontal plasticity during learning", Mendeley Data, V2, https://doi.org/10.17632/fhx5s7zxg7.2. Source data are provided with this paper.

## Code availability

Code for the data acquisition system has been made available in Github: https://github.com/ChristosLab/Wave. All code for analysis will be made available upon reasonable request

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

## Acknowledgements

Research reported in this paper was supported by the National Eye Institute of the National Institutes of Health under award number R01 EY017077 to C.C.; by NINDS training grant T32 NS073553; NIMH grant F31 MH104012 to M.R.R.; and by the Tab Williams Family Endowment. We wish to thank Kathini Palaninathan, Aquil Jones, Austin Lodish, Leonardo Silenzi, Rafael Mendoza, Macrae Robertson, and Mia Allen for technical help.

## Author contributions

C.C., M.R.R. and D.T.B. conceived and designed the experiments. M.R.R., H.T. performed the experiments. H.T., M.R.R., X.L.Q., B.S. and C.C. performed data analysis. C.C. wrote the manuscript with input from all authors.

## Competing interests

The authors declare no competing interests.
