## [Peer Review File · Nature Communications]

Prefrontal cortical plasticity during learning of cognitive tasksREVIEWER COMMENTS

Reviewer #1 (Remarks to the Author):

The authors investigate effects of neuronal population in PFC with learning, using carefully designed subsequent stages of training. They report parallel changes on a control task that just requires passive fixation, where spiking activity tends to generally go up, alpha/beta LFP activity goes down with learning. Furthermore, they nicely show a number of interesting learning effects on the active task. This is an impressive body of work, with a huge amount of data. The authors interpret the data in terms of transfer of working memory abilities, which I don't think is very convincing. However, the general topic of learning is timely, the findings are novel and interesting, the text is well written, the study is carefully executed, and the analyses are generally sound.

Major comments:

1) The general story of the manuscript is about transfer of working memory abilities. However, the design of the tasks does not seem suited to answer this. The transfer is supposed to go to a passive version of the same task. First of all, no stimulus-specific working memory seems to be required for this task, as the only requirement is to fixate. The only condition where there could be transfer would be in the case where the animals would implicitly perform the task even during passive fixation. However, the task requirements would then be basically the same, so how can you then speak of transfer from one task to another?

The main changes over learning for the passive task in terms of spiking activity are indicated in Figure 3e. This shows an increase in the activity evoked by the cue from pre-training to Phase I, which levels off, and then a slower increase during Phase III and IV, where even the activity during the delay becomes enhanced. However, the decoding results in Figure S7 show that there is no transfer of information to the passive task. The question therefore becomes what is changed for the passive task.

One interpretation is that arousal is generally increased. At least for one animal (MA), performance drops considerably during phase III and IV, suggesting that the animal required more attention to solve the task. And even if the performance was stable (as it seems for NI), the animals might have just learned to become more focused while in the setup.

An alternative interpretation is that the decoding method is not well executed or not sensitive enough, and different types of information can in fact be better decoded during the passive task. In that case, it will be hard to rule out that the animals are performing the task, even if they are just rewarded for fixation. The authors try to refute this option in the discussion section, but I think the authors would agree that it will be impossible to rule it out completely. In any case, there is no other task to perform except the one that is the same as the active task. So, it will be hard to argue that there is transfer from one task to another, as the task requirements would be identical.

2) Focusing the interpretation more broadly on the changes with learning would be preferred in my view. There are enough results that would make it into a very interesting paper. Showing learning effects in PFC in two animals trained on carefully controlled subsequent stages of a delayed-match-to-sample task is unique.

In particular, one finding that seems under-explored, is the fact that there is no response to Stimulus 1 for the active task until this stimulus becomes relevant for the task (during Phase III and in Phase IV). This is very striking in Figure 3, and even more clean in Figure 4. Although this mirrors earlier findings from the same group (Qi et al. 2011 Cerebral Cortex), the current effects are much more convincing.

What would be relevant to support this, is to perform a decoding analysis on the position of the first stimulus, using data from the same day (so not just selecting a fixed number of units). In that way, a direct comparison can be made between the active and passive tasks, during Phase IV, as well as during the early vs the late part of Phase III. Interestingly, it seems that the peak of decoding during the passive task is higher (even while only 50 units were used for the passive task and 100 units for the active task), but that it is transient, dropping off quickly.

Figure S7 seems crucial with respect to both of these comments. Would you agree to make this into a main figure?

Minor comments:

- One page 5 it is written that Phase II is “requiring the monkey to associate the match and nonmatch conditions with the corresponding choice target”. And on page 10 “This training introduced a new type of association between reward and a cognitive abstraction, the concept of “match” and “nonmatch”.” However, in contradiction to this, it is then written, “We note that throughout stage II, the monkeys could still perform the task by essentially ignoring the cue (first stimulus), since it always appeared at the same location.” Furthermore, the data supports this second interpretation. As the animals have never been trained on a match-to-sample task, it would be odd if they would immediately represent the information at this more abstract level, if they can just use the direction of Stimulus 2 (right or left). It would therefore be good to rephrase on page 5 and 10, and not use the terms match and non-match conditions.

- Similarly, for the description of Phase I, it is written, “[the animals] had to indicate if they appeared at the same or different locations by selecting one of two choice targets signifying match or nonmatch.” Again, this is misleading, it would be good to rephrase to avoid any misunderstanding.

- And similarly in Figure S7, it is suggested that the distinction between match and nonmatch conditions can be decoded from Phase II. This is again misleading, I would suggest to take it out. Furthermore, even during Phase III and IV, it is possible that the decoding of match/nonmatch is dominated by trials from a certain pair of location (for example the left-right pair that was trained extensively in Phases I-III). To rule out that it's not just the position of the stimulus that is decoded, can you show the match/nonmatch results separately for the 4 different pairs of stimulus locations?

- Furthermore, it is unclear how direction is decoded in Figure S7c. Stimulus 1 and 2 can be in different locations. Are trials taken where both of them are in the same location? It might make more sense to split the figure for Stimulus 1 and 2, so that all trials can be used.
- No significance is given for Figure S7, please add this.
- Can you add the pretraining results to Figure 3d, as well as to Figure S7a,c?
- Please show the effects of Figure 3 for the two animals separately (not just figure D, in Figure S3). It is normal for results to differ between animals, but it would be preferred to be transparent about it, in particular as the behavior seems to differ quite substantially between them.
- In Figure 1d-e, there is also a dashed square in the middle, and on page 5 it is written “appearing at a 3 × 3 grid (Fig. 1d)”. However, in the method section, it is written: “The first stimulus appeared at a different, followed by a second stimulus at the same location, or its diametric.”, suggesting that a stimulus was never presented at the center location. And also from the description of the decoding of direction results it can be concluded that there were only 8 possible directions. Can you resolve this confusion?
- Can you indicate the order in which the different directions were learned in Phase III? Was this the same for the 2 animals?
- In Figure 1i-j, maybe add a line on where the sulci run, to make it easier to see? Also, maybe orient the images in a similar way?
- Most of the information in Figure 2a-b is superfluous. The only effect that seems to be significant is the decrease in alpha and beta with learning. As alpha / beta generally shows the opposite effect as spiking activity, this finding does not add much to the story. Furthermore, the effects are hard to interpret as it could just be a decrease in signal quality. This might be unlikely, as the spiking activity increases with training (which can be noted explicitly in the text). It would be better to normalize the activity between phases, as is done for the spiking activity. However, the changes are throughout the trial, so this seems hard to do. (Maybe using the period before fixation starts?) I would therefore suggest that instead of starting with these results, add them as a final figure (mainly for EEG/MEG community). Furthermore, I would suggest to put the full spectrograms in the Supplementary Information, as it adds a lot of redundant information that is hard to make sense of.
- The legends ‘Cue-fixation’ and ‘Delay-fixation’ in many of the figures are confusing. The animals were required to fixate during the complete trial, so what it is unclear what period is meant by this. Maybe write ‘Fixation baseline’ instead? Similarly, in Figure 3c,f maybe better write ‘Fixation baseline’. Furthermore, it’s unclear which delay is meant with ‘Delay’, delay 1, delay 2, or both averaged together? I assume it’s the first delay, as this is the delay where Stimulus 1 should be kept in memory. Please clarify.
- Figure S5 is very confusing. The data indicated in panel a seems to be from the passive task. But then panel b suggests that it is data from the active task. The RFs are indicated on top of the active task, next to the line with a color as in panel a. What is also confusing is that the RF’s are only on one of the time periods during a trial. Maybe just remove panel b completely and explain in the text?
- Please add what kind of array was implanted, the brand, model, as well as length of the electrodes.

- On page 11 and 12, 'Figure 3d' is written where it seems that it should have been 'Figure 3a'.

Reviewer #3 (Remarks to the Author):

Tang and colleagues studied primate DLPFC activity over the course of training a difficult working memory task and concurrent performance of a passive viewing task. They examined the hypothesis that learned improvements in one working memory task can generalize to better functioning in other tasks. They found that over training, there were changes in DLPFC LFPs and unit discharge activity. Specifically, they found decreased beta power, increased gamma power, and increased task-related unit spiking that all also appeared during passive viewing task performance. They conclude that these neural mechanisms provide a basis for why cognitive improvements have been reported to transfer across tasks. On the whole, this is very nicely designed study, with a compelling behavioral approach, that is appropriately analyzed and addresses specific hypothesis. I believe with a few stylistic changes it will make a nice addition to the literature.

Generally, I found the figures disappointing. First, most of the stats reported in the text were not shown with data plots. I think these would help the reader follow along better with the main points. I also thought that much of the material in the supplemental figures, could be incorporated into panels of the main text figures. While reading the text, the authors kept referring back to previous figures, which required this reader to jump all around the manuscript in order to follow along. While I understand why this was necessary, it was distracting and took away from the points the authors were making. Find a way to minimize this.

In general, the results section did a good job of explaining the rationale behind each analysis, but there were too few methodological details incorporated into the results. This lead me to jump back and forth between different parts of the manuscript, even to discern basic aspects of the analysis that was currently being presented.

It's not clear to me which unit analyses were performed on the isolated units or the MUA units. This needs to be clearly defined. I would also like to see more analyses of the individual units. The hypothesis is that underlying changes in neural connectivity occur over learning and thus, this is why transfer occurs. Why wasn't this specifically tested with correlational analyses of recorded units. With such a large number of cells recorded, hopefully, there is enough data to permit such investigations.

I found Supp Fig 7 to be quite compelling. What does it mean there are unit discharge increases and less beta power if the ensemble information doesn't seem to change over learning? This should be dealt with in the Discussion.

Since the main thrust of the paper is the changes that occur over training, it is strange that there wasn't more done to show within training block changes. While the early vs. late training analysis touched on this idea, a session to session account would make for a stronger argument that the changes are indeed learning-related. An alternate hypothesis is that increasing task complexity induces different DLPFC activity patterns. The authors could address these points by showing changes within training blocks and also showing how these ephys effects relate to performance overall.

Lastly, the Discussion section is quite brief. These findings should be better explained with more detailed discussion of how they fit with other working memory and multi-task recording or imaging studies.

Response to Reviewers

We are grateful to the reviewers for their careful reading of our manuscript, their insightful comments, and their overall positive evaluation. Some common concerns were present in both reviews. We have performed extensive additional work to address all issues raised by the reviewers (*in italics below*). This work led us to draw additional insights, and we have qualified some of our conclusions. We believe the manuscript is much stronger as a result.

Reviewer #1 (Remarks to the Author):

The authors investigate effects of neuronal population in PFC with learning, using carefully designed subsequent stages of training. They report parallel changes on a control task that just requires passive fixation, where spiking activity tends to generally go up, alpha/beta LFP activity goes down with learning. Furthermore, they nicely show a number of interesting learning effects on the active task. This is an impressive body of work, with a huge amount of data. The authors interpret the data in terms of transfer of working memory abilities, which I don't think is very convincing. However, the general topic of learning is timely, the findings are novel and interesting, the text is well written, the study is carefully executed, and the analyses are generally sound.

Response: We appreciate the reviewer's kind comments. Although we believe our results are very relevant to the plasticity of the prefrontal network that allows transfer, we have accepted the reviewer's recommendation and now frame our results in a manner that emphasizes the effects of learning (see below).

Major comments:

1) The general story of the manuscript is about transfer of working memory abilities. However, the design of the tasks does not seem suited to answer this. The transfer is supposed to go to a passive version of the same task. First of all, no stimulus-specific working memory seems to be required for this task, as the only requirement is to fixate. The only condition where there could be transfer would be in the case where the animals would implicitly perform the task even during passive fixation. However, the task requirements would then be basically the same, so how can you then speak of transfer from one task to another?

The main changes over learning for the passive task in terms of spiking activity are indicated in Figure 3e. This shows an increase in the activity evoked by the cue from pre-training to Phase I, which levels off, and then a slower increase during Phase III and IV, where even the activity during the delay becomes enhanced. However, the decoding results in Figure S7 show that there is no transfer of information to the passive task. The question therefore becomes what is changed for the passive task.

One interpretation is that arousal is generally increased. At least for one animal (MA), performance drops considerably during phase III and IV, suggesting that the animal required more attention to solve the task. And even if the performance was stable (as it seems for NI), the animals might have just learned to become more focused while in the setup.

An alternative interpretation is that the decoding method is not well executed or not sensitive

enough, and different types of information can in fact be better decoded during the passive task. In that case, it will be hard to rule out that the animals are performing the task, even if they are just rewarded for fixation. The authors try to refute this option in the discussion section, but I think the authors would agree that it will be impossible to rule it out completely. In any case, there is no other task to perform except the one that is the same as the active task. So, it will be hard to argue that there is transfer from one task to another, as the task requirements would be identical.

Response: The reviewer's points are well taken. First, we have deemphasized the idea of transfer between the active and passive tasks and changed the title of the article; we now frame our results as lasting changes related to learning. Secondly, we now acknowledge that arousal may have played a role in the increase in activity in both the active and passive tasks (although we still argue that changes of activity in the passive task are unlikely to be due to arousal, alone). Thirdly, we have refined our decoding analysis (new Figure 5) and show that, after training, not only firing rate but location decoding improves in the passive task. However, decoding of the status of a stimulus as a match or a nonmatch remains much more pronounced in the active task, throughout training. Finally, we qualify our conclusions to suggest that training induces some long-lasting changes that can be probed with the passive task, however execution of the active task is characterized by unique information that neurons represent dynamically in the course of a trial.

2) Focusing the interpretation more broadly on the changes with learning would be preferred in my view. There are enough results that would make it into a very interesting paper. Showing learning effects in PFC in two animals trained on carefully controlled subsequent stages of a delayed-match-to sample task is unique.

In particular, one finding that seems under-explored, is the fact that there is no response to Stimulus 1 for the active task until this stimulus becomes relevant for the task (during Phase III and in Phase IV). This is very striking in Figure 3, and even more clean in Figure 4. Although this mirrors earlier findings from the same group (Qi et al. 2011 Cerebral Cortex), the current effects are much more convincing.

What would be relevant to support this, is to perform a decoding analysis on the position of the first stimulus, using data from the same day (so not just selecting a fixed number of units). In that way, a direct comparison can be made between the active and passive tasks, during Phase IV, as well as during the early vs the late part of Phase III. Interestingly, it seems that the peak of decoding during the passive task is higher (even while only 50 units were used for the passive task and 100 units for the active task), but that it is transient, dropping off quickly.

Figure S7 seems crucial with respect to both of these comments. Would you agree to make this into a main figure?

Response: These were excellent comments. We have recast our findings in the context of learning and changed the title of the manuscript accordingly.

The reviewer is exactly right that the response to the initial visual stimulus is very low – however there is an important caveat: during phase I, the first stimulus always appeared at the same location, and in fact ipsilateral to the recording site for one of the animals (which was a conscious design choice for us to track responses in both hemispheres – but in retrospect limited our ability to analyze the trajectory of these stimulus responses). In contrast, during phase III, the

stimulus location varied, providing an opportunity for many more units to be driven by the first stimulus. We explain that now.

We have expanded the decoding analysis. Both reviewers commented on Figure S7; we have expanded it and made this a main figure (Fig. 5). We have also included the analysis the reviewer suggests, relying on the exact same units between the passive and active tasks (Fig. 5c,f). This shows more dramatically the differences in decoding of match/nonmatch status between tasks, however the peak of stimulus-location between passive and active tasks is virtually identical, when the same neurons are used. Unfortunately, decoding of stimulus location information in the early and late stages of phase III cannot readily be compared, because different number of stimuli were presented at these times. We clarify that in the text now.

Minor comments:

1. *One page 5 it is written that Phase II is “requiring the monkey to associate the match and nonmatch conditions with the corresponding choice target”. And on page 10 “This training introduced a new type of association between reward and a cognitive abstraction, the concept of “match” and “nonmatch”.” However, in contradiction to this, it is then written, “We note that throughout stage II, the monkeys could still perform the task by essentially ignoring the cue (first stimulus), since it always appeared at the same location.” Furthermore, the data supports this second interpretation. As the animals have never been trained on a match-to-sample task, it would be odd if they would immediately represent the information at this more abstract level, if they can just use the direction of Stimulus 2 (right or left). It would therefore be good to rephrase on page 5 and 10, and not use the terms match and non-match conditions.*

Response: The reviewer points were well taken. We have revised accordingly.

2. *Similarly, for the description of Phase I, it is written, “[the animals] had to indicate if they appeared at the same or different locations by selecting one of two choice targets signifying match or nonmatch.” Again, this is misleading, it would be good to rephrase to avoid any misunderstanding.*

Response: We have revised accordingly.

3. *And similarly in Figure S7, it is suggested that the distinction between match and nonmatch conditions can be decoded from Phase II. This is again misleading, I would suggest to take it out. Furthermore, even during Phase III and IV, it is possible that the decoding of match/nonmatch is dominated by trials from a certain pair of location (for example the left-right pair that was trained extensively in Phases I-III). To rule out that it’s not just the position of the stimulus that is decoded, can you show the match/nonmatch results separately for the 4 different pairs of stimulus locations?*

Response: We wish to thank the reviewer for these excellent suggestions. Firstly, we acknowledge that the decoding of match and nonmatch information in early phases may represent anticipated responses to some extent, since these were presented in blocks. Secondly, we have performed the analysis the reviewer suggests and added a supplementary figure to break

down decoding results for different stimulus locations (Fig. S6). We show that decoding performance is not driven solely from the left-right pair.

4. Furthermore, it is unclear how direction is decoded in Figure S7c. Stimulus 1 and 2 can be in different locations. Are trials taken where both of them are in the same location? It might make more sense to split the figure for Stimulus 1 and 2, so that all trials can be used.

Response: We now present the results of the decoding analysis in much greater detail (new fig. 5). Results from only the first stimulus (cue) are shown in this figure. We have added a figure relying on an alternative analysis to quantify selectivity, based on Analysis of Variance, and show results from both first and second stimulus location (new Fig. S7).

5. No significance is given for Figure S7, please add this.

Response: Thank you for pointing this out. We have revised this figure (now Fig. 5) to show time points in which decoding performance significantly exceeded chance.

6. Can you add the pretraining results to Figure 3d, as well as to Figure S7a,c?

Response: This was a very good suggestion. We have now added these results.

7. Please show the effects of Figure 3 for the two animals separately (not just figure D, in Figure S3). It is normal for results to differ between animals, but it would be preferred to be transparent about it, in particular as the behavior seems to differ quite substantially between them.

Response: The reviewer's point is well taken. We now show these results in Fig. S1. Although some differences were present between animals, as the reviewer predicted, the beginning of training in the active task resulted in an abrupt increase in firing rate in the passive task for both subjects.

8. In Figure 1d-e, there is also a dashed square in the middle, and on page 5 it is written "appearing at a 3 × 3 grid (Fig. 1d)". However, in the method section, it is written: "The first stimulus appeared at a different, followed by a second stimulus at the same location, or its diametric.", suggesting that a stimulus was never presented at the center location. And also from the description of the decoding of direction results it can be concluded that there were only 8 possible directions. Can you resolve this confusion?

Response: We now clarify that 9 locations were used as cues but the nonmatch stimulus could only appear at 8 of those locations (the center cue stimulus was followed by a nonmatch at the rightward location). For this reason, the match/nonmatch decoding relied only on the 8 peripheral locations.

9. Can you indicate the order in which the different directions were learned in Phase III? Was this the same for the 2 animals?

Response: We now indicate the order of stimulus presentation in the methods.

10. In Figure 1i-j, maybe add a line on where the sulci run, to make it easier to see? Also, maybe orient the images in a similar way?

Response: This was an excellent suggestion. We have drawn a line through sulci. Unfortunately it is not possible to orient the images in the same way, as they were in opposite hemispheres.

11. Most of the information in Figure 2a-b is superfluous. The only effect that seems to be significant is the decrease in alpha and beta with learning. As alpha / beta generally shows the opposite effect as spiking activity, this finding does not add much to the story. Furthermore, the effects are hard to interpret as it could just be a decrease in signal quality. This might be unlikely, as the spiking activity increases with training (which can be noted explicitly in the text). It would be better to normalize the activity between phases, as is done for the spiking activity. However, the changes are throughout the trial, so this seems hard to do. (Maybe using the period before fixation starts?) I would therefore suggest that instead of starting with these results, add them as a final figure (mainly for EEG/MEG community). Furthermore, I would suggest to put the full spectrograms in the Supplementary Information, as it adds a lot of redundant information that is hard to make sense of.

Response: We appreciate the reviewer's suggestions. We have moved this figure last (Fig. 7) and eliminated the full spectrograms from the main figure (they can be appreciated in the two supplementary figures). We also clarify that we had in fact computed our results in the fashion the reviewer suggested: to make LFP power comparable across phases, we always express results relative to the baseline period of each trial. We explain that better now.

12. The legends 'Cue-fixation' and 'Delay-fixation' in many of the figures are confusing. The animals were required to fixate during the complete trial, so what it is unclear what period is meant by this. Maybe write 'Fixation baseline' instead? Similarly, in Figure 3c,f maybe better write 'Fixation baseline'. Furthermore, it's unclear which delay is meant with 'Delay', delay 1, delay 2, or both averaged together? I assume it's the first delay, as this is the delay where Stimulus 1 should be kept in memory. Please clarify.

Response: We do see the confusion these legends created. These labels meant to express that the firing rate plotted is that of the cue period minus the fixation period. We have revised to make that clearer now.

12. Figure S5 is very confusing. The data indicated in panel a seems to be from the passive task. But then panel b suggests that it is data from the active task. The RFs are indicated on top of the active task, next to the line with a color as in panel a. What is also confusing is that the RF's are

only on one of the time periods during a trial. Maybe just remove panel b completely and explain in the text?

Response: We do see the reviewer's point. All of the data were recorded from the passive task. We have eliminated panel b, which confused more than it helped, and explain in simpler terms in the text.

14. Please add what kind of array was implanted, the brand, model, as well as length of the electrodes.

Response: We now provide more details about the array in the Methods. This was a custom-made array, developed by co-author D.T. Blake.

15. On page 11 and 12, 'Figure 3d' is written where it seems that it should have been 'Figure 3a'.

Response: Thank you. We have corrected as suggested.

Reviewer #3 (Remarks to the Author):

Tang and colleagues studied primate DLPFC activity over the course of training a difficult working memory task and concurrent performance of a passive viewing task. They examined the hypothesis that learned improvements in one working memory task can generalize to better functioning in other tasks. They found that over training, there were changes in DLPFC LFPs and unit discharge activity. Specifically, they found decreased beta power, increased gamma power, and increased task-related unit spiking that all also appeared during passive viewing task performance. They conclude that these neural mechanisms provide a basis for why cognitive improvements have been reported to transfer across tasks. On the whole, this is very nicely designed study, with a compelling behavioral approach, that is appropriately analyzed and addresses specific hypothesis. I believe with a few stylistic changes it will make a nice addition to the literature.

Response: Thank you. We appreciated the kind comments and we have revised the article as suggested.

1. Generally, I found the figures disappointing. First, most of the stats reported in the text were not shown with data plots. I think these would help the reader follow along better with the main points. I also thought that much of the material in the supplemental figures, could be incorporated into panels of the main text figures. While reading the text, the authors kept referring back to previous figures, which required this reader to jump all around the manuscript in order to follow along. While I understand why this was necessary, it was distracting and took away from the points the authors were making. Find a way to minimize this.

Response: The reviewer's points are well taken. We have expanded and reorganized the figures, and moved several supplemental graphs to the main figures. We have also reordered the

references to figures to minimize the reader having to go back and forth. Different panels of Figures 3 and 4 are still presented in parallel because a merged figure would have been too busy, but all other figures are referenced in sequence.

2. In general, the results section did a good job of explaining the rationale behind each analysis, but there were too few methodological details incorporated into the results. This led me to jump back and forth between different parts of the manuscript, even to discern basic aspects of the analysis that was currently being presented.

Response: The reviewer is right that the results section was very terse. We have expanded the narrative and provided more methodological details along the presentation of the results.

3. It's not clear to me which unit analyses were performed on the isolated units or the MUA units. This needs to be clearly defined. I would also like to see more analyses of the individual units. The hypothesis is that underlying changes in neural connectivity occur over learning and thus, this is why transfer occurs. Why wasn't this specifically tested with correlational analyses of recorded units. With such a large number of cells recorded, hopefully, there is enough data to permit such investigations.

Response: We now clarify in the text and figure legends whether single units or MUA units were used in each analysis: the analysis of Fig. 3 and 6 were based on single units; the analysis of Fig. 4 and 5 was based on MUA. We appreciated the reviewer's suggestion of a correlation analysis to understand the changes in the underlying connectivity. We have added this analysis in Figure 6. We relied on spike-count correlation (also known as noise-correlation), computed during the fixation interval across all trials. Consistent with prior studies, our results show that noise correlation generally decreases after training, however the decrease was most pronounced after phase 3 and 4. Also consistent with prior studies, we show that noise correlation is lower during the active than the passive task, however values in the passive task followed the same trajectory of changes as they did in the active task.

4. I found Supp Fig 7 to be quite compelling. What does it mean there are unit discharge increases and less beta power if the ensemble information doesn't seem to change over learning? This should be dealt with in the Discussion.

Response: The reviewer's point is well taken. Reviewer 1 also raised a similar issue (point 1). In response, we have refined and extended this analysis and have now made this a main figure (Fig. 5). We show that decoding of stimulus location in fact improves in both the active and passive tasks as training proceeds. However, decoding of the status of a stimulus as a match or a nonmatch remains much more pronounced in the active task, throughout training. We therefore qualify our conclusions to suggest that training induces some long-lasting changes that can be probed with the passive task, however execution of the active task is characterized by unique information that neurons represent dynamically in the course of a trial. We discuss in more detail in the Discussion section.

5. Since the main thrust of the paper is the changes that occur over training, it is strange that there wasn't more done to show within training block changes. While the early vs. late training analysis touched on this idea, a session to session account would make for a stronger argument that the changes are indeed learning-related. An alternate hypothesis is that increasing task complexity induces different DLPFC activity patterns. The authors could address these points by showing changes within training blocks and also showing how these ephys effects relate to performance overall.

Response: This was an excellent suggestion. We have added this analysis in Figure S4.

6. Lastly, the Discussion section is quite brief. These findings should be better explained with more detailed discussion of how they fit with other working memory and multi-task recording or imaging studies.

Response: We have now expanded the discussion considerably.

REVIEWERS' COMMENTS

Reviewer #1 (Remarks to the Author):

The authors did an impressive job on improving their manuscript, with a number of additional analyses and a restructuring of the flow of the paper which has substantially improved it. I only have a few minor points about the new results.

- Figure S1 is a very useful addition. If I understand correctly, monkey NI had it's recording chamber ipsilateral to the initial cue location, which explains the increase in the response to the cue during phase III for that monkey? If so, it would be good to explicitly mention this in the text.
- It is very surprising that match/non-match can be decoded during the passive task during a number of time periods, even though the traces are overlapping with the shuffled data. The authors seem to suggest this could be due to a block effect. Couldn't the authors control for this, e.g. by cross-validating across blocks?
- Dashed lines are used in panels 5a,b, d, e for shuffled data, but for the passive condition in panels 5c, f. This is confusing. Maybe change into e.g. thick and a thin lines for panels 5c, f?
- Could significance tests be added to Figure 5c, f?
- About the significance tests of the decoding analysis it is written in the method section: "The actual data with a mean outside the 95% confidence interval of the shuffled data was confided showed a significant response." I'm not sure I follow this. Would it be possible to rephrase?

Reviewer #3 (Remarks to the Author):

The authors have sufficiently addressed all my concerns from the previous version, they substantially improved this manuscript and I now believe it is ready for publication. I especially appreciate how the authors have now clearly delineated what does and what doesn't transfer between tasks over learning. I think that this opens a ripe new area for future exploration and greatly advances our understanding of learning induced network changes.

NCOMMS-20-41606B

Tang et al.

Response to Reviewers

We are grateful to the reviewers for their positive evaluation. We also appreciate their remaining comments, and attention to detail, which have helped us to fine tune the manuscript.

Reviewer #1 (Remarks to the Author):

The authors did an impressive job on improving their manuscript, with a number of additional analyses and a restructuring of the flow of the paper which has substantially improved it. I only have a few minor points about the new results.

- Figure S1 is a very useful addition. If I understand correctly, monkey NI had it's recording chamber ipsilateral to the initial cue location, which explains the increase in the response to the cue during phase III for that monkey? If so, it would be good to explicitly mention this in the text.

Response: We now clarify that cue presentation was ipsilateral to the chronic array for one of the two monkeys, though few single neurons from either monkey had receptive fields that corresponded to the initial cue location in the active task, in phases I and II (Fig. 1a-b). However, both animals exhibited an increase in responses in the passive task after training (for the preferred location of each single neuron), a result which is informative about the effects of training in a task, that are not only evident for stimuli that were used in the active task.

- It is very surprising that match/non-match can be decoded during the passive task during a number of time periods, even though the traces are overlapping with the shuffled data. The authors seem to suggest this could be due to a block effect. Couldn't the authors control for this, e.g. by cross-validating across blocks?

Response: We are grateful to the reviewer for spotting this issue. For some of the conditions in the initial permutation test, the size of the pseudo-populations sampled was close to the total number of neurons available in the condition and the variability generated by sampling in this fashion was artifactually small, resulting in many false-positive decoding results. This only happened in the passive task, as many more responsive MUAs were recorded in the active task. We have now repeated our permutation analysis, using always 200 neurons, sampled 500 times for each test. We have revised Figure 5 and the Methods section accordingly.

- Dashed lines are used in panels 5a,b, d, e for shuffled data, but for the passive condition in panels 5c, f. This is confusing. Maybe change into e.g. thick and a thin lines for panels 5c, f?

Response: The reviewer's point is well taken. We have implemented this change in Figure 5.

- Could significance tests be added to Figure 5c, f?

Response: The significance tests have been added.

- About the significance tests of the decoding analysis it is written in the method section: "The actual data with a mean outside the 95% confidence interval of the shuffled data was confided showed a significant response." I'm not sure I follow this. Would it be possible to rephrase?

Response: We have rephrased the description of the decoding analysis (which was revised, as noted above)

Reviewer #3 (Remarks to the Author):

The authors have sufficiently addressed all my concerns from the previous version, they substantially improved this manuscript and I now believe it is ready for publication. I especially appreciate how the authors have now clearly delineated what does and what doesn't transfer between tasks over learning. I think that this opens a ripe new area for future exploration and greatly advances our understanding of learning induced network changes.

Response: We appreciate the reviewer's comments.

Reporting Requirements

Some additional edits have been made in the manuscript to report sample sizes in the figure legends and statistics more fully, as dictated by the Journal's Reporting Requirements.